# Evolution of Thyroglobulin Loop Kinetics in EpCAM

**DOI:** 10.3390/life11090915

**Published:** 2021-09-03

**Authors:** Serena H. Chen, David R. Bell

**Affiliations:** 1Oak Ridge National Laboratory, Computational Sciences and Engineering Division, Oak Ridge, TN 37830, USA; 2Advanced Biomedical Computational Science, Frederick National Laboratory for Cancer Research, Frederick, MD 21701, USA

**Keywords:** kinetics, EpCAM, thyroglobulin loop, molecular dynamics simulation, evolution

## Abstract

Epithelial cell-activating molecule (EpCAM) is an important cancer biomarker and therapeutic target given its elevated expression in epithelial cancers. EpCAM is a type I transmembrane protein that forms *cis*-dimers along the thyroglobulin type-1A-like domain (TYD) in the extracellular region. The thyroglobulin loop (TY loop) within the TYD is structurally dynamic in the monomer state of human EpCAM, binding reversibly to a TYD site. However, it is not known if this flexibility is prevalent across different species. Here, we conduct over 17 μs of all-atom molecular dynamics simulations to study EpCAM TY loop kinetics of five different species, including human, mouse, chicken, frog, and fish. We find that the TY loop remains dynamic across evolution. In addition to the TYD binding site, we discover a second binding site for the TY loop in the C-terminal domain (CTD). Calculations of the dissociation rate constants from the simulation trajectories suggest a differential binding pattern of fish EpCAM and other organisms. Whereas fish TY loop has comparable binding for both TYD and CTD sites, the TY loops of other species preferably bind the TYD site. A hybrid construct of fish EpCAM with human TY loop restores the TYD binding preference, suggesting robust effects of the TY loop sequence on its dynamic behavior. Our findings provide insights into the structural dynamics of EpCAM and its implication in physiological functions.

## 1. Introduction

Epithelial cell-activating molecule (EpCAM, CD326) is a cell surface protein that is a prominent cancer biomarker and therapeutic target due to its overexpression on epithelial tumors [1,2]. The extracellular region of EpCAM consists of three domains: an N-terminal domain (NTD), a thyroglobulin type-1A-like domain (TYD), and a C-terminal domain (CTD) which connects to a transmembrane helix [3,4]. EpCAM exists as a dimer, in which the identical monomers associate along the interface of their TYDs, leaving the oft-antibody targeted NTD surface exposed [3]. EpCAM function remains unknown; however, numerous interaction partners have implied several roles, including cell proliferation and differentiation [5,6] as well as various signaling pathways including regulated intramembrane proteolysis [7,8].

EpCAM’s TYD is dichotomous in that it forms EpCAM dimers with high avidity (*K_d_* < 10 nM) [9], yet it contains a known dibasic proteolytic cleavage site between Arg 80 and Arg 81, which is buried in the dimer state. Pavsic et al. [3] posited that there exists a dynamic equilibrium between monomer and dimer states which allows for proteolytic cleavage at this site. When EpCAM is cleaved, it can no longer form a dimer [3]. Interestingly, in our previous work, we found reversible opening and closing behavior of the TY loop with the TYD of human EpCAM in the monomer state [10]. However, it is not clear if the dynamic behavior of the TY loop affects EpCAM dimerization or exists throughout evolution. Based on a multiple sequence alignment, the TY loop and dibasic cleavage sites are relatively well conserved across evolutionary disparate species, where they are thought to behave similarly for dimerization and cleavage [3]. The exception is fish EpCAM, which does not have a dibasic cleavage site in the TY loop. Yet, EpCAM remains critical to fish development and normal functioning [11,12]. Ultimately, our understanding of EpCAM evolutionary sequence effects remains limited.

Here, we seek to elucidate the evolution of EpCAM TY loop kinetics from human to other organisms and probe how sequence affects the reversible opening and closing behavior of the TY loop as observed in human EpCAM. We explore TY loop dynamics using molecular dynamics (MD) simulations, which have been used successfully to study intrinsically disordered proteins [13,14] as well as predict binding properties of sequence mutations [15,16]. From extensive all-atom MD simulations, we compute rate constants of TY loop dynamics using the direct counting method [17,18,19], and we find that the TY loop remains flexible across all organisms. In addition to binding the TYD as previously studied, we discover a second binding site, located in the CTD and away from the putative location of the membrane. Binding of the TY loop to this CTD site has a higher dissociation rate constant (*k_off_*) than to the TYD site for all organisms except for fish, whose TY loop is five residues longer than the other organisms studied. Further investigation of EpCAM-TY loop hybrids reveals that the different TY loop sequences are responsible for the differential CTD binding behavior between fish and human. Our work holds important implications for understanding EpCAM evolution and physiology.

## 2. Methods

In this work, we built seven all-atom EpCAM models for human (*Homo sapiens,* Uniprot ID: P16422), mouse (*Mus musculus,* Uniprot ID: Q99JW5), chicken (*Gallus gallus,* Uniprot ID: Q5F381), frog (*Xenopus laevis,* Uniprot ID: Q8AWG0), fish (*Danio rerio,* Uniprot ID: Q6DRJ5), and two hybrid systems, human EpCAM with fish TY loop (human + FTY) and fish EpCAM with human TY loop (fish + HTY). All models were built starting from the X-ray crystal structure of human EpCAM (PDB ID: 4MZV [3]). Three experimentally engineered Gln (Gln 74, 111, and 198) were reverted to Asn according to the Uniprot sequence and the natural Met 115 variant was used. All models were built using MODELLER [20].

The resulting all-atom models were each solvated in the center of a water box with a minimum distance of 15 Å from the edge of the water box to the nearest protein atom. The systems were first neutralized, and then ionized to a salt concentration of 0.15 M using Na^+^ Cl^−^ ions. Six disulfide bonds were added between Cys 27 and 46, 29 and 59, 38 and 48, 66 and 99, 110 and 116, and 118 and 135. Solvation, ionization, and addition of disulfide bonds were performed with VMD [21]. Following a similar protocol to our previous studies [22,23], the resulting systems were each subjected to 20,000 steps of energy minimization with all heavy atoms in the protein fixed, another 20,000 steps of energy minimization with all atoms free to move, followed by 250 ps equilibration with a 0.5 fs time step. After equilibration, a 500 ns trajectory was generated in a production run with a 2.0 fs time step. For each system, five independent 500 ns trajectories were performed. Structures were taken every 200 ps for analysis, yielding a total of 12,500 structures for each system. All MD simulations were performed with NAMD 2.13 [24] in the NPT ensemble at 1 atm and 310 K. The CHARMM36m force fields [25] and TIP3P water model [26] were used. The nonbonding interactions were calculated with a typical cutoff distance of 12 Å, while the long-range electrostatic interactions were enumerated with the Particle Mesh Ewald algorithm [27].

## 3. Results

The MD simulations of human EpCAM reveal the high structural flexibility of its TY loop. To characterize the TY loop dynamics, we calculated the contact ratio of the TY loop with EpCAM based on the number of structures that a contact is present to the total of 12,500 MD structures (Figure 1A). We define a contact as the distance between the geometric center of all heavy atoms of the TY loop and each EpCAM residue using cutoff distances of 6.5, 8.0, and 12.0 Å. In addition to the region that is near the TY loop in primary sequence, we identified two sites with relatively high contact ratio. One site is between position 109 and 115 in the TYD, which corresponds to the binding site previously observed to form reversible binding behavior with the TY loop [10]. The other binding site is between position 219 and 225 in the CTD, which is a new discovery, and we refer it to as the CTD site. The contact ratio of the TY loop to the CTD site is lower than that to the TYD site but higher than that to the rest of the EpCAM. To further explore the binding pattern of the TY loop residues, we calculated the residue-residue contact ratio of the TY loop with EpCAM (Figure 1B). We found that different parts of the TY loop interact with the two sites, where residues 84 to 94 of the TY loop bind the TYD site and residues 78 to 82 bind the CTD site. Together, these contact analyses suggest three distinct states of human EpCAM: (1) the TY loop in the open conformation, (2) the TY loop binding to the TYD site (TYD_closed_), and (3) the TY loop binding to the CTD site (CTD_closed_). Figure 1C shows the crystal structure of the EpCAM monomer, which is an open conformation. Two EpCAM monomers in the open conformation form a *cis* dimer along their TY loop interfaces [3]. Figure 1D shows the TY loop binding the TYD site by ‘flipping down’ towards the membrane. Similarly, Figure 1E shows the TY loop binding the CTD site by ‘flipping up’ away from the membrane. Notably, we observed only transient interactions between the TY loop and the epitope-rich N-terminal domain (NTD, residues 24 to 62) of EpCAM. Although many antibodies bind the NTD, the orientation of the TY loop does not allow easy binding access to the NTD.

It is not known if the dynamic binding behavior of the TY loop is unique to human EpCAM. Therefore, we explored the structural dynamics for EpCAMs of other organisms, including mouse, chicken, frog, and fish. We built homology models based on a multiple sequence alignment of their EpCAM protein sequences [3] and performed MD simulations as we conducted for human EpCAM. Figure 2A presents alignments of these sequences at the TY loop, TYD site, and CTD site. Overall, the sequences are well conserved at these regions, except for fish TY loop, which has additional five residues and thus longer compared to the TY loops of other organisms (Figure 2B). From their MD structures, we again calculated the contact ratio of the TY loops with corresponding EpCAMs (Figure 2C). Interestingly, the TY loops of these organisms also have higher contact ratios to the residues at the TYD and CTD sites, suggesting that this dynamic binding behavior of the TY loop is prevalent across evolution.

Despite high sequence similarity of the TY loop, TYD site, and CTD site between organisms, we questioned whether TY loop binding kinetics were preserved throughout evolution. Therefore, we calculated the rate constants of the TY loop binding to the TYD and CTD sites. Based on human TY loop and human EpCAM residue contact analysis (Appendix A), we considered an EpCAM structure is in the TYD_closed_ form when there are at least three TY loop residues are in contact with the TYD site. Similarly, a structure is in the CTD_closed_ form if there are at least three TY loop residues contacting the CTD site. We computed the dissociation rate constant ($k_{off}$) and the association rate constant ($k_{on}$) according to the direct counting method [17,18,19]:(1)$$k_{off}=\frac{1}{mean\;survival\;time}$$
(2)$$k_{on}=\frac{1}{t_{on}}v_{eff}N_{Av}$$
where the *mean survival time* is the average survival time of the bound state, $t_{on}$ is the average unbound time, $v_{eff}$ is the effective volume (simulation box volume—EpCAM volume) [18], and $N_{Av}$ is Avogadro’s number. Figure 3A shows $k_{off}$, which depicts how quickly the TY loop unbinds. For most organisms including human, the TY loop has a lower $k_{off}$ for the TYD site than the CTD site, suggesting that TYD site binding is stronger given similar $k_{on}$ values shown in Figure 3B. The exception to this finding is fish, where the $k_{off}$ values for the TYD and CTD sites are comparable. The fish TY loop is distinct in that it has five extra residues in the TY loop (26 residues total) than all other organisms (21 residues total). To determine if differences in the TY loop sequences were driving this binding behavior, we built models for two hybrid systems: fish EpCAM sequence with human TY loop (fish + HTY) as well as human EpCAM with fish TY loop (human + FTY), and we performed the same MD simulations and analysis on these hybrid constructs. We found that, unlike the fish system, the fish + HTY hybrid system has higher $k_{off}$ for the CTD site than the TYD site, similar to other organisms. On the other hand, human + FTY system resulted in overlapping $k_{off}$ values for the CTD and TYD sites similar to the fish system. This result indicates that the TY loop sequence drives the binding kinetics for the two sites. All organisms exhibit similar $k_{on}$ values, including the hybrid systems, suggesting that all systems have similar propensity for the open form.

It is interesting that by switching only the TY loop sequences, we can restore the different binding kinetics for the CTD site of the fish and human systems. To further explore how the CTD residues of fish EpCAM interacts with the fish and human TY loops, we computed the contact ratio of each residue at the CTD site for the fish and fish + HTY systems (Figure 3C). The two lysine residues (K221 and K223) of the CTD site have the highest contact ratio with the TY loops. Moreover, except for K221, all other CTD residues have lower contact ratios in the hybrid system than in the fish system. We also computed the contact ratio of the TY loop residues to find out which TY loop residues interact with the CTD site (Appendix A). Both fish and human TY loops have a net charge of +2; however, fish TY loop has more charged residues than human TY loop. The negatively charged residue triplet, D85, E86 and N87, of the fish TY loop closely binds and forms large contacts with the CTD site (Figure 3D and Appendix A), indicating that in the fish system the binding of TY loop for the CTD site is driven by specific electrostatic and charged-polar interactions. For the fish + HTY system, residues E81, L84, N86 and N87 have the highest contact ratios with the CTD site, yet we observe the binding interaction is driven by only one salt bridge between E81 and the two lysine residues as well as hydrophobic interactions (Figure 3E). This result agrees with the $k_{off}$ values, which supports that the fish TY loop binds more strongly to the CTD site than the human TY loop.

## 4. Discussion

Here, we characterized EpCAM TY loop kinetics across evolutionary species using all-atom MD simulations. In addition to the previously identified TYD binding site, we discovered a second TY loop binding site in the CTD that is present in all organisms studied. From computing rate constants, we find that the TYD binding site is favored over the CTD binding site for all organisms except for fish, where TYD and CTD binding are competitive. From studying fish/human hybrid sequences, we determine that the longer, divergent fish TY loop sequence is responsible for this different binding behavior.

Though many studies have modeled EpCAM [3,10,28,29,30,31], we have not found reported evidence of the TY loop-CTD binding site. This is understandable given that these studies have mostly focused on predicting EpCAM interactions with therapeutics targeting the physiological *cis*-dimer form [10,29,31]. Likewise, researchers have focused on characterizing other EpCAM domains such as the intramembrane [3] and intracellular [30] regions. Žagar et al. [28] report modeling wild-type and mutant extracellular EpCAM dimers to probe key interacting residues of dimer formation and identify three mutations at the TY loop (K83D, P84D, L88D) that result in large changes in predicted solvation free energy of dimer formation. MD simulations confirmed that these mutations drastically destabilized the EpCAM dimer interface. Similarly, Gaber et al. [32] conducted short (10 ns) MD simulations to elucidate mass spectroscopy data and characterize EpCAM oligomerization structure.

One differential aspect of our work is long-timescale modeling of EpCAM monomer rather than the crystallized and physiological EpCAM *cis*-dimer form. Our decision to model the monomer form of EpCAM was driven by several factors. First, in our previous work, we only observed reversible binding behavior of the TY loop in the monomer form; we did not observe TY loop flexibility when it was in the dimer form. Second, the reported crystal structure work [3] as well as several subsequent works [28,33] have shown that the EpCAM dimer must separate for physiological cleavage sites to be accessible to proteases. The exact mechanism and extent of this separation has not been determined; however, a dynamic equilibrium between dimer and monomer states has been proposed [3]. This implies that although EpCAM forms *cis* dimers with high affinity (*K_D_* < 10 nM), EpCAM monomers do occur in vivo. Lastly, we sought to characterize evolutionary kinetics of the TY loop domain. Although the TY loop domain is similar in most organisms, it is distinct in fish, being five residues longer than other organisms. This extension in fish, combined with the disordered nature of the TY loop as well as the lack of non-human EpCAM experimental structures prevented us from modeling evolutionary EpCAM dimer structures with high confidence before understanding the TY loop dynamics in their monomer states.

An important note about our work is the use of a consistent temperature across different organisms. For all MD simulations studied here, we used a temperature of 310 K corresponding to human body temperature. Mouse and chicken body temperatures are in the range of 309.5 K–311 K and 313.6 K–314.7 K, respectively, which deviates slightly from our simulated temperature. Fish (*Danio rerio*) and frog (*Xenopus laevis*) are cold blooded, meaning their body temperatures will equilibrate to their environment, which is typically closer to room temperature, although *Danio rerio* can tolerate temperatures up to 314.7 K [34]. Different temperatures, acting via thermal fluctuations of atoms and molecules, may affect the kinetics of the TY loop in ways that are not readily apparent, including possible temperature-modulated affects for fish and frog. However, this is beyond the scope of this work. Here, under the same temperature at 310 K, our simulation results reveal different binding preferences of the TY loop between fish and the other organisms.

We speculate about several possible physiological roles of the EpCAM CTD binding site. We note that when the TY loop is bound to the CTD site, EpCAM is effectively prevented from forming dimers along the known TYD dimerization interface. This interface is not occluded when the TY loop interacts with the TYD site, even though the TY loop is not in the correct open configuration to dimerize. We also note that when the TY loop is contacting the CTD site, the ‘flipped up’ loop conformation may make the GRR proteolytic cleavage site of residues 79 to 81 more accessible to extracellular proteases. We analyzed the trajectories for conformational changes in this GRR proteolytic site when the TY loop bound the CTD site, but we did not observe distinct trends in GRR’s solvent exposed surface area nor backbone conformation. Interestingly, when the TY loop binds the CTD site, the NTD largely remains uncontacted by the TY loop, indicating that NTD epitopes remain surface exposed for effector binding.

One concern is whether the CTD or TYD binding site is occluded by glycans. Known glycosylated residues N74, N111, and N198 are not located near the CTD site residues 219 to 225 in human EpCAM. Specifically, N198 is on the opposite side of EpCAM as the TY loop and is therefore not thought to have a large effect on the TY loop. In the dimer form, the N74Q mutant of the solved crystal structure is pointed laterally away from the TY loop and is not thought to play a role in dimerization [3]. In the monomer form, N74 is at the start of the TY loop and therefore may have some effect on TY loop dynamics. However, it is several residues away from the TY loop residues 78–82 and 84–94, which have the largest contact with the CTD and TYD sites, respectively. N111 is likely to have a more prominent effect on TY loop kinetics due to its location on the TYD binding site (residues 109–115). N-linked glycans are often found to shield protein sites from interaction [35,36], and that could be the purpose of N111 for EpCAM, to shield the TYD site from interacting with the TY loop. If this is the role of the N111 glycan, then it is interesting to note that the CTD binding site does not have a shielding glycan, possibly implicating a specialized purpose for CTD binding. Here, we did not model EpCAM glycans due to several reasons. We know from a previous work [37] that human EpCAM’s glycans consist of two hybrid oligosaccharides and one high mannose chain. However, we do not know the exact composition nor location of each glycan. Compounding this further is that even less is known about EpCAM glycosylation in other organisms. Some sites, such as N111, are preserved across species [3], while other sites, such as N74, are recent additions in primates. Given our paucity of data on EpCAM glycosylation, there is not a straightforward way to unambiguously compare TY loop kinetics across species with glycosylated EpCAM. Hence, here, we decided to leave out EpCAM glycans for our comparison of TY loop kinetics. Ultimately, our study demonstrates that the TY loop sequence determines the binding kinetics for the CTD site, and that CTD site binding will likely disrupt EpCAM dimerization, though to what physiological effect remains to be elucidated.

## Figures and Tables

**Figure 1 life-11-00915-f001:**
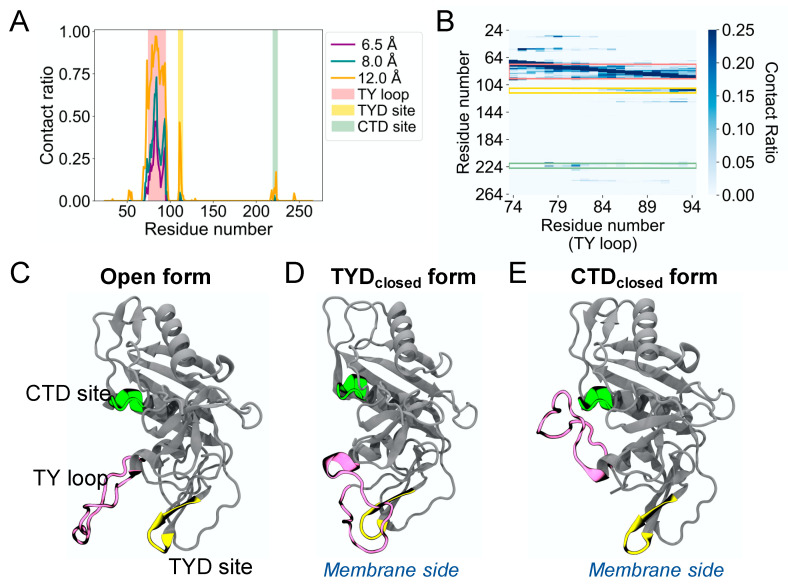
Molecular dynamics simulations reveal a second binding site in the CTD for the TY loop of human EpCAM. (**A**) Contact ratio of the TY loop with EpCAM. Contact ratio is calculated as the ratio of the number of structures that a contact is present to the total number of MD structures. A contact is defined as the distance between the geometric center of all heavy atoms of the TY loop and each EpCAM residue based on three cutoff distances: 6.5 Å (purple), 8.0 Å (dark green), and 12.0 Å (orange). The residues of the TY loop, the TYD binding site (TYD site), and the CTD binding site (CTD site) are highlighted in pink, yellow, and light green, respectively. (**B**) Heat map of residue-residue contact ratio of the TY loop with EpCAM. A residue pair is in contact if any atoms are within 4.5 Å. (**C**–**E**) Three-dimensional structures of three states of EpCAM: (**C**) Open, (**D**) the TY loop binding to the TYD site (TYD_closed_), and (**E**) the TY loop binding to the CTD site (CTD_closed_). The TY loop, TYD site, and CTD site are color coded as in A and B. The location of the membrane is shown when EpCAM is membrane bound.

**Figure 2 life-11-00915-f002:**
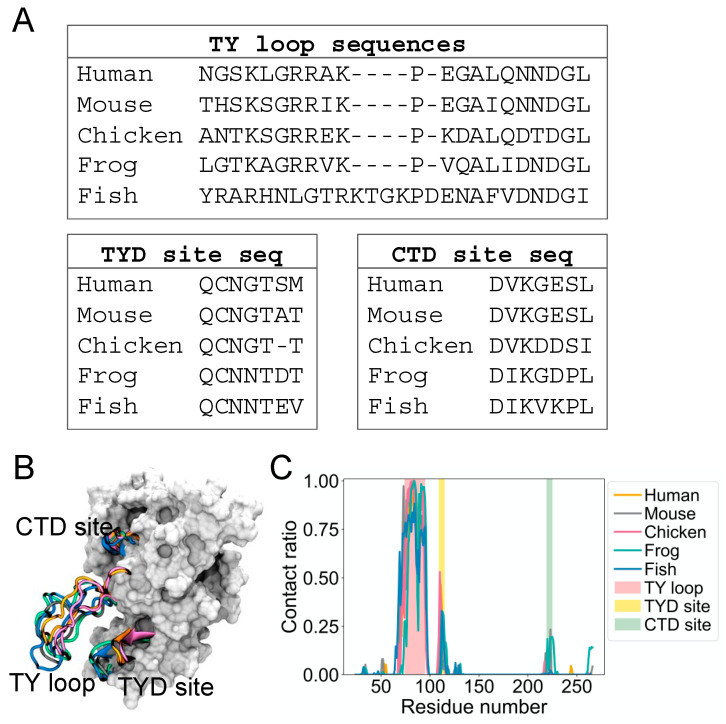
The binding site in the CTD for the TY loop is present in other species. (**A**) Sequence alignment of the TY loop, TYD site, and CTD site for EpCAMs of five species: human, mouse, chicken, frog, and fish. (**B**) Overlapped initial EpCAM models of human (orange), mouse (gray), chicken (magenta), frog (green), and fish (blue). The TY loop, TYD site, and CTD site of five species are in cartoon while the rest of human EpCAM is in surface representation for reference. (**C**) Contact ratio of the TY loop with EpCAM of each species. A contact is defined as in Figure 1A, where the cutoff distance is 12 Å.

**Figure 3 life-11-00915-f003:**
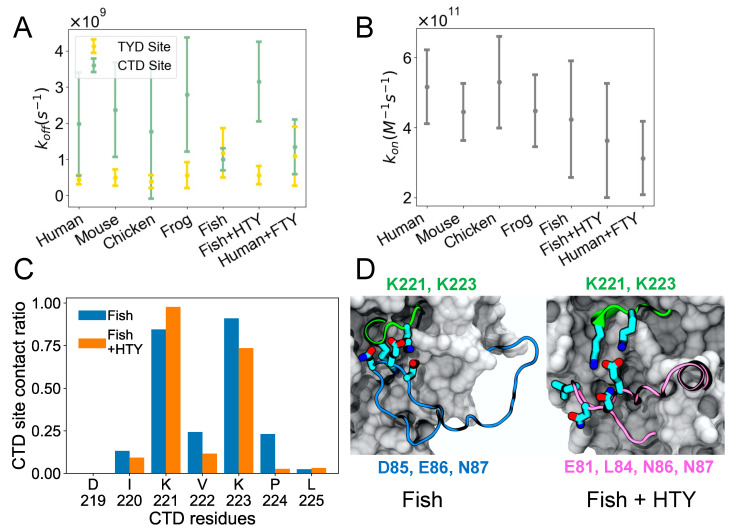
Binding kinetics of TY loop and details of the CTD site. (**A**) Dissociation rate constant (*k_off_*) of TY loop for TYD and CTD sites across evolutionary species as well as hybrid systems. Fish + HTY stands for fish EpCAM with human TY loop, while human + FTY stands for human EpCAM with fish TY loop. (**B**) Association rate constant (*k_on_*) of TY loop. (**C**) Contact ratio of each residue at the CTD site for the fish and fish + HTY systems. The contact ratio is defined as the number of structures that a residue at the CTD site is in contact with the TY loop to the total number of structures in the CTD_closed_ form. A residue pair is in contact if any atoms are within 4.5 Å. Residue numbers are from the fish EpCAM sequence. (**D**) Representative CTD_closed_ forms for fish (left) and fish + HTY (right) systems. CTD site residues K221 and K223 have large contact ratios for both systems. However, note that except for K221, all other CTD residues have lower contact ratios in the hybrid system than in the fish system. Key interacting residues of the two TY loops are shown below the images in blue and pink, respectively. All error bars in A and B represent 95% confidence intervals.

## Data Availability

All data supporting the findings of this study are described in the Methods and Results and are available from the corresponding authors upon reasonable request.

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
