# Peer review of "Evolution of Thyroglobulin Loop Kinetics in EpCAM"

_life, 2021, doi:10.3390/life11090915_

Round 1

Reviewer 1 Report

Authors investigated  the loop dynamics in EpCAM protein. Interestingly, they found that the loop can adopt different conformations corresponding to both open and closed states. Further, authors compared the loop dynamics of the protein in four different species. The work appears interesting to general readers and MD simulations were carefully performed. Therefore, I recommend it for publication after a minor revision.

Authors used the same temperature in simulations of the proteins in four different species. Do authors think that 310K might be too high for the body temperature of a fish? Authors might want to state that the same temperature is solely used for statistical sampling.   

Reviewer 2 Report

Dear authors,

I have read and analyzed the manuscript entitled "Evolution of Thyroglobulin Loop Kinetics in EpCAM", submitted by S. H. Chen and D. R. Bell, to be considered for publication in MDPI Life.

The work described in the manuscript touches an important cell-surface carcinoma marker EpCAM which is present on cell surface (basically) as a dimer. Several proteolytic enzymes cleave it at various sites within the extracellular region, and these cleavages are implicated in initiation of nuclear signaling via the cleaved-off cytosolic tail of EpCAM. Since in the dimeric form (some) cleavage sites are not accessible, EpCAM dimer must at least temporarily dissociate for these sites to become accessible, therefore consideration of EpCAM in monomeric form is necessary.
In the work presented the authors tackle exactly this topic—what is the conformational dynamics of EpCAM as monomer? They focus on extracellular region of EpCAM, specifically on the loop of the thyroglobulin domain (TYD) which is in the dimeric form involved in subunit–subunit interactions, however in the monomeric form it is exposed and seems conformationally flexible. In a previous work (co-authored by D. R. Bell, https://doi.org/10.1073/pnas.1913242117) it has already been demonstrated (using molecular dynamics simulations) that the loop of TYD adopts some preferential conformations, and in this work the authors describe this in more detail.
By performing molecular dynamics simulations of extracellular part of human EpCAM as a monomer (taken from crystal structure of the dimer), and of homology structural models of equivalent parts of EpCAM from other species, the authors provide these main conclusions: (1) demonstrate that TYD loop is flexible in EpCAM from all selected organisms; (3) show that there are two interaction sites for TYD loop, one I TYD itself and the other within the C-terminal domain; (3) highlight that conformational dynamics TYD loop in fish EpCAM differs from the other organisms. In fish TYD loop is 5 residues longer and does not have a dibasic site (Arg–Arg, matriptase cleavage site) that is characteristic for other organisms.

The methods used are appropriate and well described. Also, the results obtained are presented in a clear and understandable manner, supported by appropriate figures. The choice of organisms is appropriate since it includes human EpCAM (most studied), murine EpCAM (often used as model organism), plus fish, amphibian and bird EpCAM. The results presented and briefly listed give a unique and interesting insight into EpCAM evolution—structural dynamics relationship.

I have no objections for publishing the manuscript, expect a request to address EpCAM glycosylation more thoroughly. Authors already provide a very brief mentioning of near the end of discussion. The two N-glycosylation sites that are proximal to the TYD loop are N74 and N111. The N111 is part of the site where the TYD loop interacts with the rest of the TYD (residue stretch 109–115, listed 
